# Are Tumor Marker Tests Applied Appropriately in Clinical Practice? A Healthcare Claims Data Analysis

**DOI:** 10.3390/diagnostics13213379

**Published:** 2023-11-03

**Authors:** Sabrina M. Stollberg, Markus Näpflin, Michael Nagler, Carola A. Huber

**Affiliations:** 1Department of Health Sciences, Helsana Insurance Group, 8081 Zurich, Switzerland; markus.naepflin@helsana.ch (M.N.); carola.huber@helsana.ch (C.A.H.); 2Department of Clinical Chemistry, Inselspital, Bern University Hospital, University of Bern, 3010 Bern, Switzerland; michael.nagler@insel.ch; 3Institute of Primary Care, University Hospital Zurich, University of Zurich, 8091 Zurich, Switzerland

**Keywords:** tumor marker determination, healthcare claims data, guideline adherence, inappropriateness, laboratory testing, overutilization

## Abstract

Tumor markers (TM) are crucial in the monitoring of cancer treatment. However, inappropriate requests for screening reasons have a high risk of false positive and negative findings, which can lead to patient anxiety and unnecessary follow-up examinations. We aimed to assess the appropriateness of TM testing in outpatient practice in Switzerland. We conducted a retrospective cohort study based on healthcare claims data. Patients who had received at least one out of seven TM tests (CEA, CA19-9, CA125, CA15-3, CA72-4, Calcitonin, or NSE) between 2018 and 2021 were analyzed. Appropriate determinations were defined as a request with a corresponding cancer-related diagnosis or intervention. Appropriateness of TM determination by patient characteristics and prescriber specialty was estimated by using multivariate analyses. A total of 51,395 TM determinations in 36,537 patients were included. An amount of 41.6% of all TM were determined appropriately. General practitioners most often determined TM (44.3%) and had the lowest number of appropriate requests (27.8%). A strong predictor for appropriate determinations were requests by medical oncologists. A remarkable proportion of TM testing was performed inappropriately, particularly in the primary care setting. Our results suggest that a considerable proportion of the population is at risk for various harms associated with misinterpretations of TM test results.

## 1. Introduction

Tumor markers (TM) have a crucial role in managing various types of cancer. They are utilized to estimate the prognosis, monitor the efficacy of treatment, and detect relapse early [1,2]. However, because of their limited sensitivity and specificity, most TM are not appropriate to be used as screening parameters or to clarify non-specific clinical findings [3,4,5]. Unjustified determinations may not only limit laboratory resources and increase unnecessary costs for the healthcare system, but can also harm the patient by leading to unnecessary further investigations, interventions, and patient anxiety in case of false positive results. Major medical guidelines have recognized the risks associated with over testing and oppose TM use as diagnostic tool for cancer detection [6,7,8,9,10,11,12,13,14,15,16,17,18,19,20,21,22,23]. A few, mostly outdated, studies conducted in different healthcare settings indicated that TM are frequently misused as screening parameters [24,25,26]. The number of appropriate determinations varies considerably with figures as low as five percent [25]. It is, however, unknown how TM are used in routine clinical practice in Switzerland. In addition, little is known about predictors of appropriate requests on both the physician and patient sides. Therefore, we aimed to, first, determine the frequency of a predefined set of TM, second, determine the proportion of TM appropriately applied as defined by a laboratory test with a respective cancer-related disease and intervention (CDI) and, third, investigate the appropriateness of TM determination according to the prescriber specialty and patient characteristics using a large Swiss claims database.

## 2. Material and Methods

### 2.1. Study Design and Setting

We conducted a retrospective cohort study using healthcare claims data from the Helsana Health Insurance Group (Helsana). Helsana is one of the largest health insurance companies in Switzerland, covering an average of around 1,395,000 mandatory insured patients from all parts of the country, which corresponds to around 15% of the Swiss population. Linked at the patient level and based on healthcare invoices for reimbursement, the database includes longitudinal information about patients’ sociodemographics, medications prescribed, laboratory tests received, use of outpatient and inpatient healthcare, and associated costs. The content of the obligatory health insurance in Switzerland is regulated by law. Therefore, there are no regulatory differences between Helsana and other Swiss health insurances companies. Several previous studies focusing on different contexts strongly suggest the included study population (Helsana insurance collective) to be representative of the Swiss population and validated the high quality and completeness of the insurance’s database [27,28,29].

We studied persons aged 18 years or older with mandatory health insurance at Helsana between 2018 and 2021, who had received at least one TM test in the given years. The first determination of any TM in this period was set as the patient’s index date. Additional TM determinations after the index date or within the look-back period were not included in further analysis. To determine whether the TM determination was appropriate, the five years preceding and the following 12 months after the index date were examined for a CDI (antineoplastics, diagnosis, radiooncological therapy, in- or outpatient operations, histology, CT-thorax-abdomen for cancer staging, and two or more visits to an oncologist). Patients were excluded if they were not insured with Helsana during the entire look-back and follow-up period. Patients were classified under the term “Cancer diagnosis (CD) reliable” if they had any of the following CDI: prescription of antineoplastic agents, a diagnosis related to the single TM or a radio oncological therapy. Approximation codes for “CD probable” were an in- or outpatient surgical procedure probably associated with cancer, e.g., a rectal resection for rectal cancer. Approximation codes for “CD conceivable” were codes for histology, radiological procedures associated with cancer staging, namely CT-thorax and abdomen contemporaneous, or two or more visits to the oncologist. Patients with none of the listed codes were classified as “No CD”. The categories “CD reliable” and “CD probable” were considered as “appropriate request”, and “CD conceivable” and “no CD” as “inappropriate request”. Appendix A lists all CDI assigned to the related TM. Appendix A shows the flow chart and classification of CDI.

### 2.2. Testing of TM

The TM tests were selected from the list of analyses (AL) from the Federal Office of Public Health [30]. The AL contains all laboratory tests which are covered by the mandatory health insurance in Switzerland. The markers were selected due to their broad use and their (in most cases) unique assignment to a cancer entity.

The following TM tests were included in the analysis according evidence-based guidelines for appropriate testing: Carbohydrate antigen 15-3 (CA15-3) for breast cancer [12,13,14], Carbohydrate antigen 19-9 (CA19-9) for pancreatic and biliary tract cancer [8,31,32], Carbohydrate antigen 72-4 (CA72-4) for gastric cancer [9,21], Carbohydrate antigen 125 (CA125) for ovarian cancer [7,33,34], Carcinoembryonic antigen (CEA) for colorectal cancer [10,18,20,23,35,36], neuron-specific enolase (NSE) for lung cancer [6,22,37,38,39], and Calcitonin for medullary thyroid cancer [18,35,36,40]. Appendix A provides detailed information on each TM and its associated medical guidelines.

### 2.3. Statistical Analysis

Descriptive statistics were calculated for the patient characteristics, proportion and appropriateness of TM use, cost analysis and physicians’ specialization. The direct laboratory costs for the TM were taken from the health insurance bills of each patient. The costs are shown in Swiss francs as well as in Euro and US dollar to enable international comparison. The currency conversion was based on the annual average rate given by the Swiss tax authorities [41].

Appropriateness of TM testing was examined with a binomial regression model with “logit” link function. Statistical significance was set at the 0.05 level.

All analyses were performed using the statistical program R, version 4.2.2 (R Foundation for Statistical Computing, Vienna, Austria).

## 3. Results

### 3.1. Patient Characteristics

Table 1 summarizes patient characteristics for all patients with at least one TM determination between 2018 and 2021. Patients with TM determinations were mostly female (25,170, 68.9%), mainly enrolled in a managed care program (20,073, 54.9%), had a mean age of 64.5 years and were mostly residents of the German-speaking area of Switzerland (22,865, 62.6%).

### 3.2. Frequency of TM Determinations

In total, 205,160 determinations of TM were detected in 51,780 patients between 2018 and 2021 in the Helsana population. After exclusion of patients who were not continuously insured for the total time of the look-back and follow-up period and only counting the first TM determination per patient (index date), 51,395 determinations of TM were detected in 36,537 patients between 2018 and 2021.

The most frequently determined test between 2018 and 2021 was CEA for colorectal cancer (22,583, 43.9% of all TM), followed by CA125 (9082, 17.7%), CA15-3 (8118, 15.8%), CA19-9 (7672, 14.9%), Calcitonin (1414, 2.8%), NSE (1303, 2.5%) and CA72-4 (1223, 2.4%), respectively (Table 2).

### 3.3. Number of TM Determinations per Patient at Index Date

Analysis showed that most patients had one TM determined at the index date (25,728, 70.4%), 7838 patients (21.5%) had two different TM, and 2971 (8.1%) had three or more TM determined. The mean number of TM determined in one patient at index date was 1.41 (Appendix A).

### 3.4. Detected CDI and Appropriateness of TM Requests

For all TM combined, 28.5% (10,425) of patients received antineoplastic medication, most frequently those in which the TM CA15-3 for breast cancer was determined (4301, 53.0%) and least frequently the ones with Calcitonin for medullary thyroid cancer (153, 10.8%). An average of 21.0% (7669) of the patients had a predefined ICD-10 diagnosis, again most of the patients with the TM CA15-3 (2881, 35.5%). An intervention billed using a predefined outpatient operation code was only rarely detected in 0.7% during the look-back or follow-up period of the single TM (243 patients), whereas a predefined inpatient operation code was found in 20.8% (7594) of all patients. A code for histology was found in 71.0% (25,939) of all patients. 21.0% (7682) of all patients had no evidence of cancer (Table 2).

The TM requests of 12,631 patients (34.6%) were classified as “CD reliable” and in 2542 (7.0%) patients they were classified as “CD probable”, so that in 15,173 patients (41.6%) the TM determination was classified as “appropriate”. CA15-3 was the marker that was most often appropriately requested (4532, 55.9%), followed by NSE (476, 36.5%), CEA (7948, 35.2%), CA125 (3082, 33.9%), CA19-9 (2214, 32.9%), Calcitonin (434, 30.7%), and CA72-4 (262, 21.5%), respectively (Table 3).

### 3.5. Cost Analysis

The total cost for all 36,537 TM were CHF 1,205,842 (EUR 1,096,220, USD 1,262,661). The proportion of costs for appropriately determined TM was 36.5%, corresponding to CHF 440,370 (EUR 400,336, USD 461,120).

### 3.6. TM Determination According to Physicians’ Specialization

Among the specialists, general practitioners (GP) most frequently requested at least one of the seven TM (16,199, 44.3%), followed by the group “others” (5843, 16.0%) and gynecologists (5048, 13.8%) (Table 4).

The marker most frequently determined by GP was CEA for colorectal cancer (12,550, 55.6%), as was the one with the “others” group (3947, 17.5%). Gynecologists determined CA125 most frequently (4206, 46.3%) but ranked also top 3 for CA72-4 determination, which they determined in 214 patients (17.5%). CA15-3 was most frequently determined by GP and not by gynecologists (2976, 36.7%).

### 3.7. Multivariate Analysis on Appropriateness

Table 5 shows the multivariate analysis for appropriateness for each TM. Strong predictors for appropriate determinations across all individual TM were requested by oncologists or tertiary hospitals. For example, oncologists determined CEA in reference to GP appropriately with an odds ratio of 11.2 (95% confidence interval (CI) 9.93, 12.6, *p* < 0.001). Language region (French and Italian) was in most TM associated with a risk for less appropriate determination, whereas type of deductible was not. Age was a slightly positive predictor for appropriate determination in six out of the seven analyzed TM.

## 4. Discussion

To the best of our knowledge, the present study is the first to evaluate appropriateness of TM determination in a large cohort based on healthcare claims data, and additionally to analyze patient and prescriber characteristics, as well as the first to evaluate TM determinations in Switzerland.

The study reveals the following key results:

First, only 41.6% of all TM determinations were classified as appropriate requests. The mean number of different TM requests at a time was 1.41. These findings suggest that a substantial proportion of determinations are made as part of screening tests or to clarify non-specific clinical findings.

Although comparability poses difficulties, e.g., due to different markers examined in different study populations as well as various underlying guidelines, the results are still in line with previous, mostly older, studies:

For example, Ntaios et al. analyzed TM requests retrospectively in their hospital and found a total of 9782 inappropriate TM orders in a ten-month period during 2008; for the TM CA125, AFP, CA19-9, CYFRA21-1 and NSE, the adequate requests were under 10% [42]. Moreno et al. [43] analyzed laboratory requests from the University Hospital of Padua. In the two-year study period between 2011 and 2013, 23,059 analytical requests of TM were analyzed, and 39.9% were classified as appropriate. The mean number of TM requested was 2.4 and 26.6% of requests ordered four or more TM at a time. Arioli et al. [25] interestingly found that only a five percent minority of TM requests was appropriate in their department of Internal Medicine in the Hospital in Modena. A more recent study from 2020 revealed similar results: In a teaching hospital only 12.9% of TM requests had an underlying cancer diagnosis [44]. Studies in the outpatient setting also showed comparable results. For example, in a study published in 2017, Gion et al., analyzed electronic health records of a Local Health Authority and found that 59.2% of the 52,536 outpatients for whom a TM was ordered were without a cancer code. A mean of 1.54 TM per person was ordered [45].

Second, most TM determinations in the outpatient setting were requested by GP (16,199, 44.3%), followed by the group “Others” (5843, 16.0%), gynecologists (5048, 13.8%), and tertiary hospitals (4582, 8.8%). This result is of particular interest since there are only very few studies examining partially this issue. For example, a Brazilian retrospective analysis based on healthcare claims data from 2010 to 2017 examined the medical specialty of ordering physicians and found that, interestingly, most of the physicians were cardiologists (23.9%) [46]. However, numbers of insured patients were rather low and only 1112 TM tests were analyzed in the whole period. In our study, GP least frequently ordered TM adequately (27.8%). Most appropriate determinations were requested by medical oncologists (77.3%). Age had a statistically significant but very small effect. Interestingly, we detected 5874 (16.1%) of TM requests in patients older than 79 years. Despite the increased life expectancy, an inadequate TM determination at this age seems even more questionable due to the often lacking (therapeutical) consequences. Guideline based screening interventions such as coloscopy or gynecological pap smear have regularly set age limits. Thus, screening or clarifying non-specific findings using TM, which are not evidence-based, indicate a high degree of inappropriateness in these elderly patients. Moreover, we detected 155 male patients with a CA125 determination, a marker for ovarian cancer, that is 1.7% of all CA125 requests. Although, some cases might be explained by inadvertent and accidental requests, there are previous findings showing that up to 33% of patients with CA125 determinations are of male gender [47,48]. This large group of inappropriate requests could be caused by using lab block testing as a screening tool. The rare diagnoses that might justify a determination in male patients, e.g., para testicular papillary carcinoma [49], probably cannot explain all of these requests. Laboratory forms with ready-made block orders are especially questionable in these cases. Schulenburg-Brand et al. [47] have—in addition to training measures—set up a laboratory information system that automatically rejected CA125 requests in male patients and found an absolute decrease from 127 to 27 requests.

It could be explained that there are clinical cases where a TM determination is useful even without a coded CDI, such as paraneoplastic syndrome. However, these cases are very rare on the one hand and cannot account for the huge number of inadequate TM determinations, on the other hand [50].

Third, the total costs for index-date TM determinations between 2018 and 2021 were CHF 1,205,842 (EUR 1,096,220, USD 1,262,661). The proportion of costs for appropriately determined TM was 36.8%. Considering a health–economic perspective: The total healthcare costs in Switzerland were CHF 82,472 Mio (EUR 74,299 Mio, USD 83,305 Mio) in 2019 [51]. Although the cost for inappropriate TM determination might appear negligible, it should be emphasized that these are only the costs of the laboratory tests, and the costs for possible follow-up interventions are not quantified. Zhang et al. [52] found that inappropriate TM requests accounted for 1.3% to 2.1% of their hospitalization costs. Ntaios et al. [42] found that the total absolute cost for inappropriate TM testing over a 10-month period at their large hospital was EUR 239,748.

Remarkably, the potential damage to patients and their—unneeded—anxiety cannot be quantified. Moreno et al. [43] found that a remarkable 43% of the patients who had a positive result of the TM determinations had no cancer diagnosis, i.e., had a false positive result. For this reason, it is important to create awareness of the damage caused by medical over- and misutilization: Ntaios et al. [42] have therefore thought about a different term for the so-called “TM”; they suggested changing it to “tumor progression markers”.

## 5. Strengths and Limitations

The present study has some limitations that need to be considered: First, despite the in general systematic coding of CDI, misclassification of CDI cannot completely be excluded. In Switzerland, physicians do not need to code the patient’s diagnosis in the outpatient setting. Therefore, for patients whose cancer diagnosis was not made in a hospital, an ICD-10 diagnosis is not available. Second, no data on TM determination in hospitalized patients exists in our healthcare claims data, since laboratory analyses in hospitalized patients are billed by case flat rates. Third, our classification of the TM requests as appropriate or inappropriate is partly rather broad and more defined in guidelines. For example, CA15-3 determination in a patient with localized breast cancer after cancer diagnosis or for surveillance would be classified as appropriate in our analysis, but according to guidelines only metastasized patients should have a determination of CA15-3 [53]. Further, oncological-defined ICD-O3 diagnoses are not available in Helsana Group healthcare claims data. Therefore, no specification is possible for the morphological diagnosis of cancer, e.g., medullary thyroid cancer or neuroendocrine carcinoma. Additionally, the billing code for histology represents a very broad approximation parameter: The histological result could be benign or malign, it could have been caused by a dermatological excision or a visceral operation. We therefore classified a proven code under “CD conceivable”. Fourth, our data do not include the outcome of the TM determination (e.g., positive, negative, false positive, and false negative). Thus, we cannot measure, for example, follow-up cost due to false positive markers and subsequent investigations. Data collected from hospitals and medical diagnostic centers might offer additional information about the clinical course. However, data are not comprehensive and available nationwide in Switzerland. Further research would be valuable, if both data sources (healthcare claims data and clinical data) could be linked anonymously to provide the most comprehensive information possible.

However, the present study also has several important strengths. First, it is based on a large study population covering about 1.4 million health insurance customers from all over Switzerland. Thus, the study most likely reflects the reality of daily medical routine very well and provides valuable real-world evidence. Second, we could systematically code CDI due to several reimbursement and approximation codes like antineoplastics, inpatient ICD-10 diagnosis, or operation codes. Third, we were able to analyze appropriate requests on a prescriber and patient level. Thus, possible influencing factors for inappropriate TM determinations were considered and deducted. Additionally, study findings provide a solid base for the discussion of public health strategies in order to reduce further inappropriate TM determinations.

## 6. Conclusions

According to the present study, inappropriate determination of TM is a major problem in routine medical care, particularly in the primary care setting. Our results suggest a considerable proportion of the population at risk for various harms associated with misinterpretations of TM test results. Efforts to increase awareness among healthcare providers and patients about the potential harm of TM determinations are needed.

## Figures and Tables

**Table 1 diagnostics-13-03379-t001:** Characteristics of the cohort of patients with tumor marker (TM) determination.

Characteristic	All TM, *N* = 36,537 ^a^ (%)
**Gender**	
Women	25,170 (68.9%)
Men	11,367 (31.1%)
**Age ^b^**	67 (54, 76)
**Age group**	
<20	88 (0.2%)
20–29	715 (2.0%)
30–39	1499 (4.1%)
40–49	3858 (10.6%)
50–59	6518 (17.8%)
60–69	8192 (22.4%)
70–79	9793 (26.8%)
>79	5874 (16.1%)
**Managed care**	20,073 (54.9%)
**Language region**	
German	22,865 (62.6%)
French	8178 (22.4%)
Italian	5494 (15.0%)

^a^ Number, percent; ^b^ Median (interquartile range (IQR)).

**Table 2 diagnostics-13-03379-t002:** Cancer-related diseases and interventions (CDI) in the previous 5 years and following 1 year after index tumor marker (TM) determination.

			Appropriate Request							Inappropriate Request				
	Main Application Field (Cancer)	Total	CD ^a^ Reliable *N* (%)	Antineo-Plastics *N* (%)	Diagnoses *N* (%)	Radio Oncology *N* (%)	CD ^a^ Probable *N* (%)	Outpatient Operations *N* (%)	Inpatient Operations *N* (%)	CD ^a^ Conceivable *N* (%)	Histology *N* (%)	CT Thorax Abdomen *N* (%)	>2 Visits to Oncologist *N* (%)	No CD ^a^ *N* (%)
**All TM ^b^**		36,537	12,631 (34.6%)	10,425 (28.5%)	7669 (21.0%)	4369 (12.0%)	7794 (21.3%)	243 (0.7%)	7594 (20.8%)	27,636 (75.6%)	25,939 (71.0%)	9980 (27.3%)	9235 (25.3%)	7682 (21.0%)
**CA 125**	Ovarian	9082	2028 (22.3%)	1844 (20.3%)	435 (4.8%)	564 (6.2%)	1636 (18.0%)	72 (0.8%)	1565 (17.2%)	6614 (72.8%)	6315 (69.5%)	1624 (17.9%)	1528 (16.8%)	2037 (22.4%)
**CA 15-3**	Breast	8118	4519 (55.7%)	4301 (53.0%)	2881 (35.5%)	2303 (28.4%)	646 (8.0%)	103 (1.3%)	557 (6.9%)	6786 (83.6%)	6238 (76.8%)	2269 (28.0%)	3946 (48.6%)	1179 (14.5%)
**CA 19-9**	Pancreatic, biliary tract	7672	2178 (28.4%)	1957 (25.5%)	457 (6.0%)	610 (7.9%)	839 (10.9%)	0 (0.0%)	839 (10.9%)	6011 (78.3%)	5628 (73.4%)	2605 (34.0%)	1807 (23.6%)	1526 (19.9%)
**CA 72-4**	Gastric	1223	255 (20.9%)	224 (18.3%)	38 (3.1%)	68 (5.6%)	31 (2.5%)	0 (0.0%)	31 (2.5%)	920 (75.2%)	872 (71.3%)	298 (24.4%)	177 (14.5%)	290 (23.7%)
**Calcitonin**	Medullary thyroid	1414	245 (17.3%)	153 (10.8%)	73 (5.2%)	53 (3.7%)	288 (20.4%)	0 (0.0%)	288 (20.4%)	1019 (72.1%)	986 (69.7%)	208 (14.7%)	75 (5.3%)	309 (21.9%)
**CEA**	Colorectal	22,583	7688 (34.0%)	6336 (28.1%)	2564 (11.4%)	2615 (11.6%)	2701 (12.0%)	3 (0.0%)	2699 (12.0%)	16,745 (74.1%)	15,675 (69.4%)	6912 (30.6%)	5727 (25.4%)	5468 (24.2%)
**NSE**	Lung	1303	472 (36.2%)	423 (32.5%)	107 (8.2%)	198 (15.2%)	43 (3.3%)	0 (0.0%)	43 (3.3%)	1148 (88.1%)	1090 (83.7%)	451 (34.6%)	402 (30.9%)	146 (11.2%)

^a^ Cancer Diagnosis. ^b^ All codes for patients with multiple positive cancer codes were listed, so that the sum of each row is more than 100%. Likewise, the sum of each column is more than the number of all TM together since patients often had more than one TM determined.

**Table 3 diagnostics-13-03379-t003:** Appropriateness of tumor marker (TM) requests.

	Appropriate Request			Inappropriate Request	
	Total	CD ^a^ Reliable *N* (%)	CD ^a^ Probable *N* (%)	CD ^a^ Conceivable *N* (%)	No CD ^a^ *N* (%)
**All TM**	36,537	12,631 (34.6%)	2542 (7.0%)	13,682 (37.4%)	7682 (21.0%)
**CA 125**	9082	2028 (22.3%)	1054 (11.6%)	3981 (43.8%)	2019 (22.2%)
**CA 15-3**	8118	4519 (55.7%)	13 (0.2%)	2407 (29.7%)	1179 (14.5%)
**CA 19-9**	7672	2178 (28.4%)	346 (4.5%)	3628 (47.3%)	1520 (19.8%)
**CA 72-4**	1223	255 (20.9%)	7 (0.6%)	672 (54.9%)	289 (23.6%)
**Calcitonin**	1414	245 (17.3%)	189 (13.4%)	672 (47.5%)	308 (21.8%)
**CEA**	22,583	7688 (34.0%)	260 (1.2%)	9218 (40.8%)	5417 (24.0%)
**NSE**	1303	472 (36.2%)	4 (0.3%)	681 (52.3%)	146 (11.2%)

**^a^** Cancer diagnosis.

**Table 4 diagnostics-13-03379-t004:** Ranking of medical specialties within the group of general practitioners (GP) and outpatient specialists.

	Total N	Top 1 ^a^	Top 2 ^a^	Top 3 ^a^	Top 4 ^a^	Top 5 ^a^	Top 6 ^a^
**All TM**	36,537	GP 16,199 (44.3%) (*27.8%*)	Others ^b^ 5843 (16.0%) (*45.0*%)	Gynecology 5048 (13.8%) (*37.3*%)	Tertiary Hospital ^c^ 4582 (12.5%) (*69.3*%)	Medical Oncology 2841 (7.8%) (*77.3*%)	Group practices ^d^ 2024 (5.5%) (*39.1*%)
**CEA**	22,583	GP 12,550 (55.6%) (*21.9%*)	Others ^b^ 3947 (17.5%) (*40.9*%)	Tertiary Hospital ^c^ 2068 (9.2%) (*72.7*%)	Medical Oncology 1782 (7.9%) (*73.8*%*)*	Group practices ^d^ 1187 (5.3%) (*31.4*%)	Gynecology 1049 (4.6%) (*37.2*%)
**CA 125**	9082	Gynecology 4206 (46.3%) (*27.7%*)	GP 2074 (22.8%) (*25.9*%)	Tertiary Hospital ^c^ 1090 (12.0%) (*55.9*%)	Others ^b^ 804 (8.9%) (*37.3*%)	Group practices ^d^ 547 (6.0%) (*38.4*%)	Medical Oncology 361 (4.0%) (*71.7*%)
**CA 15-3**	8118	GP 2976 (36.7%) (*32.7*%)	Medical Oncology 1608 (19.8%) (*78.0*%)	Gynecology 1082 (13.3%) (*60.4*%)	Others ^b^ 1059 (13.0%) (*55.1*%)	Tertiary Hospital ^c^ 858 (10.6%) (*85.4*%)	Group practices ^d^ 535 (6.6%) (*62.4*%)
**CA 72-4**	1223	GP 541 (44.2%) (*14.6*%)	Others ^b^ 224 (18.3%) (*30.8*%)	Gynecology 214 (17.5%) (*15.4*%)	Tertiary Hospital ^c^ 156 (12.8%) (*32.1*%)	Group practices ^d^ 48 (3.9%) (*16.7*%)	Medical Oncology 40 (3.3%) (*57.5*%)
**CA 19-9**	7672	GP 3561 (46.4%) (*23.1%*)	Others ^b^ 1597 (20.8%) (*31.2*%)	Tertiary Hospital ^c^ 1169 (15.2%) (*57.7*%)	Gastroenterology 472 (6.2%) (*30.1*%)	Medical Oncology 455 (5.9%) (*65.5*%)	Group practices ^d^ 418 (5.4%) (*21.5*%)
**Calcitonin**	1414	Tertiary Hospital ^c^ 483 (34.2%) (*42.4*%)	GP 388 (27.4%) (*20.9*%)	Others ^b^ 189 (13.4%) (*27.5*%)	Endocrinology 167 (11.8%) (*30.5*%)	Group practices ^d^ 134 (9.5%) (*25.4*%)	Nuclear Medicine 53 (3.7%) (*20.8*%)
**NSE**	1303	GP 348 (26.7%) (*19.8*%)	Tertiary Hospital ^c^ 345 (26.5%) (*53.6*%)	Others ^b^ 279 (21.4%) (*38.4*%)	Dermatology 167 (12.8%) (*21.6*%)	Medical Oncology 99 (7.6%) (*66.7*%)	Group practices ^d^ 65 (5.0%) (*20.0*%)

^a^ Number, percent, percent appropriate request. ^b^ Specialists ranked after the top 5 nominated ones were subsumed under “Others”. ^c^ Tertiary hospitals are central providers, in Switzerland defined by the treatment of more than 9000 inpatient cases per year and a sum of more than 20 training categories. ^d^ Group practices are defined by their organization (e.g., shared premises) and not by the specialty of the physician involved. It is not possible to deduct the specialty from our data. The service providers were categorized using the classification of the Swiss paying agent register (created by SASIS AG).

**Table 5 diagnostics-13-03379-t005:** Regression analysis on appropriateness.

	CEA			CA 125			CA 15-3			CA 72-4			CA 19-9			Calcitonin			NSE		
Characteristic	OR ^a^	95% CI ^b^	*p*-Value	OR ^a^	95% CI ^b^	*p*-Value	OR ^a^	95% CI ^b^	*p*-Value	OR ^a^	95% CI ^b^	*p*-Value	OR ^a^	95% CI ^b^	*p*-Value	OR ^a^	95% CI ^b^	*p*-Value	OR ^a^	95% CI ^b^	*p*-Value
**Gender**																					
Female	—	—		—	—		—	—		—	—		—	—		—	—		—	—	
Male	0.93	0.88, 1.0	0.034	1.09	0.73, 1.61	0.7	**0.56**	0.39, 0.80	0.002	**1.40**	1.02, 1.93	0.037	**1.30**	1.17, 1.44	<0.001	1.16	0.88, 1.53	0.3	**1.46**	1.14, 1.88	0.003
**Age**	**1.03**	1.02, 1.03	<0.001	**1.03**	1.02, 1.03	<0.001	**1.02**	1.01, 1.02	<0.001	**1.03**	1.02, 1.04	<0.001	**1.02**	1.02, 1.03	<0.001	1.00	1.00, 1.01	0.4	**1.03**	1.02, 1.04	<0.001
**Language region**																					
German	—	—		—	—		—	—		—	—		—	—		—	—		—	—	
French	**0.59**	0.55, 0.64	<0.001	**0.74**	0.66, 0.83	<0.001	**0.59**	0.52, 0.67	<0.001	0.81	0.54, 1.19	0.3	**0.68**	0.60, 0.77	<0.001	**0.60**	0.40, 0.88	0.011	0.97	0.62, 1.50	0.9
Italian	**0.59**	0.55, 0.65	<0.001	**0.54**	0.46, 0.64	<0.001	**0.46**	0.40, 0.53	<0.001	**0.48**	0.27, 0.84	0.013	**0.45**	0.39, 0.53	<0.001	0.90	0.61, 1.33	0.6	0.87	0.55, 1.36	0.5
**Area**																					
City	—	—		—	—		—	—		—	—		—	—		—	—		—	—	
Suburban	**1.13**	1.05, 1.22	0.002	**1.24**	1.11, 1.40	<0.001	**1.30**	1.14, 1.48	<0.001	0.99	0.68, 1.42	>0.9	1.12	0.98, 1.27	0.10	1.18	0.88, 1.57	0.3	1.06	0.79, 1.42	0.7
Countryside	**1.27**	1.16, 1.40	<0.001	**1.24**	1.08, 1.43	0.003	**1.22**	1.04, 1.43	0.013	1.21	0.74, 1.94	0.4	**1.24**	1.06, 1.46	0.008	**1.53**	1.08, 2.17	0.017	0.99	0.66, 1.46	>0.9
**Managed care**																					
Standard	—	—		—	—		—	—		—	—		—	—		—	—		—	—	
Managed	**1.15**	1.08, 1.22	<0.001	1.09	0.99, 1.20	0.066	**1.15**	1.04, 1.27	0.006	1.02	0.76, 1.37	0.9	1.10	0.99, 1.22	0.080	0.99	0.78, 1.26	>0.9	0.87	0.68, 1.11	0.3
**Deductible**																					
Low	—	—		—	—		—	—		—	—		—	—		—	—		—	—	
High	0.92	0.84, 1.01	0.091	0.93	0.82, 1.06	0.3	0.91	0.78, 1.06	0.2	1.00	0.65, 1.51	>0.9	0.96	0.82, 1.11	0.6	1.25	0.91, 1.70	0.2	0.75	0.52, 1.07	0.11
**Provider**																					
GP	—	—		—	—		—	—		—	—		—	—		—	—		—	—	
Group practices	**1.66**	1.45, 1.89	<0.001	**1.99**	1.62, 2.44	<0.001	**3.15**	2.59, 3.85	<0.001	1.01	0.42, 2.17	>0.9	0.84	0.65, 1.08	0.2	1.16	0.72, 1.85	0.5	1.00	0.49, 1.92	>0.9
Gynecology	**2.26**	1.97, 2.60	<0.001	**1.36**	1.20, 1.54	<0.001	**2.98**	2.57, 3.46	<0.001	1.33	0.82, 2.14	0.2	1.12	0.82, 1.50	0.5	1.29	0.27, 4.65	0.7	3.26	0.15, 36.1	0.3
Medical Oncology	**11.2**	9.93, 12.6	<0.001	**7.19**	5.59, 9.31	<0.001	**7.93**	6.86, 9.18	<0.001	**7.93**	3.96, 16.2	<0.001	**7.12**	5.75, 8.84	<0.001	**30.9**	5.24, 588	0.002	**6.89**	4.15, 11.6	<0.001
Tertiary Hospital	**10.2**	9.17, 11.4	<0.001	**3.96**	3.37, 4.65	<0.001	**13.9**	11.3, 17.2	<0.001	**2.74**	1.78, 4.19	<0.001	**4.44**	3.85, 5.13	<0.001	**2.47**	1.80, 3.41	<0.001	**4.21**	2.98, 5.98	<0.001
Others	**2.57**	2.38, 2.78	<0.001	**1.77**	1.48, 2.11	<0.001	**2.59**	2.24, 3.01	<0.001	**2.51**	1.72, 3.68	<0.001	**1.74**	1.53, 1.98	<0.001	1.34	0.96, 1.88	0.089	**1.79**	1.28, 2.53	<0.001
**Number of different TM**	1.19	0.51, 2.67	0.7	1.09	0.53, 2.20	0.8	2.04	0.47, 14.1	0.4	0.75	0.16, 2.50	0.7	1.33	0.33, 4.71	0.7	1.07	0.05, 9.30	>0.9	0.60	0.13, 2.08	0.5

^a^ OR = Odds ratio, significant results are in bold; ^b^ CI = Confidence interval.

## Data Availability

The datasets analyzed during the current study are not publicly available due to reasons of individual privacy, legal and regulatory affairs, but are available from the corresponding author on reasonable request.

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
