# Peer review of "Are Tumor Marker Tests Applied Appropriately in Clinical Practice? A Healthcare Claims Data Analysis"

_diagnostics, 2023, doi:10.3390/diagnostics13213379_

Round 1
Reviewer 1 Report
Comments and Suggestions for Authors
Dear authors,
I want to congratulate for this topic. Given the psychological impact of cancer on both patients and physicians, tumor markers related to a case study for appropriateness. In recent decades, the trends of TM recommendation has increased rapidly, although the updated national and international guidelines clearly establish the appropriate clinical scenario for requesting TM and their minimum retest interval. Within MT, we believe that effective management of inappropriate requests from the laboratory can maximize the effectiveness of these tests in the clinic with a large impact on the quality of the laboratory service, while also allowing the introduction of new and theoretically more efficient MT-s without exceeding the available budgets. I think that this article may have a good practical importance, an d can clarified the topic for the practitioners. I don't have any recommendations, except maybe the introduction, that can be improve.
Good luck!
Reviewer 2 Report
Comments and Suggestions for Authors
This paper is clearly clearly written and well organized. The introduction and background are reasonable given the reasonable premise of the manuscript. Data presented in the tables is comprehensive and helpful. few suggestions to authors.
1. The authors collected the data from Helsana insurance company. If they collected the data from hospital and diagnostic center medical records it will be more helpful for the data accuracy.
2. There are some typographical errors in the manuscript.
Comments on the Quality of English Language
Minor editing of English language required
